# Changes in sucrose metabolism in maize varieties with different cadmium sensitivities under cadmium stress

**Cong Li[1,2], Yu Liu[1], Jing Tian[1,2], Yanshu Zhu[1,2], Jinjuan Fan** [1,2] *

**1** College of Biological Science and Technology, Shenyang Agricultural University, Shenyang, Liaoning, China, **2** Shenyang Key Laboratory of Maize Genomic Selection Breeding, Shenyang, Liaoning, China

* jinjuanf@hotmail.com

**Data Availability Statement:** All relevant data are within the manuscript and its Supporting Information files.

## Abstract

Sucrose metabolism contributes to the growth and development of plants and helps plants cope with abiotic stresses, including stress from Cd. Many of these processes are not well-defined, including the mechanism underlying the response of sucrose metabolism to Cd stress. In this study, we investigated how sucrose metabolism in maize varieties with low (FY9) and high (SY33) sensitivities to Cd changed in response to different levels of Cd (0 (control), 5, 10, and 20 mg L$^{-1}$ Cd). The results showed that photosynthesis was impaired, and the biomass decreased, in both varieties of maize at different Cd concentrations. Cd inhibited the activities of sucrose phosphate synthase (SPS) and sucrose synthase (SS) (sucrose synthesis), and stimulated the activities of acid invertase (AI) and SS (sucrose hydrolysis). The total soluble sugar contents were higher in the Cd-treated seedlings than in the control. Also, Cd concentrations in the shoots were higher in SY33 than in FY9, and in the roots were lower in SY33 than in FY9. The decreases in the photosynthetic rate, synthesis of photosynthetic products, enzyme activity in sucrose synthesis direction, and increases in activity in hydrolysis direction were more obvious in SY33 (the sensitive variety) than in FY9 (the tolerant variety), and more photosynthetic products were converted into soluble sugar in SY33 than in FY9 as the Cd stress increased. The transcript levels of the sugar transporter genes also differed between the two varieties at different concentrations of Cd. These results suggest that sucrose metabolism may be a secondary response to Cd additions, and that the Cd-sensitive variety used more carbohydrates to defend against Cd stress rather than to support growth than the Cd-tolerant variety.

## Introduction

Because of industrial and agricultural activities, heavy metal pollution has become a major environmental problem worldwide [1]. Cadmium (Cd), a highly toxic heavy metal, is very soluble in water and is readily absorbed by plant roots and transported through the xylem [2, 3]. Cd in soils at high levels can limit the growth of plants and crop yields, and may also threaten human health if transferred through the food chain [4–6]. Excessive accumulation of Cd in plant cells causes biochemical and physiological changes and inhibits the growth and development of plants [7]. The

**Funding:** This research was financially supported by the scientific research projects of Liaoning Provincial Department of Education, China (No. LSNZD201608 and LSNYB201607). The funders had no role in study design, data collection and analysis, decision to publish, or preparation of the manuscript.

**Competing interests:** The authors have declared that no competing interests exist.

accumulation of Cd also stimulates the production of reactive oxygen species (ROS), such as superoxide ($O_2^-$), hydrogen peroxide ($H_2O_2$), and hydroxyl radicals ($OH^-$), which cause oxidative stress and lead to lipid peroxidation, membrane damage, and enzyme inactivation in plant cells [8–10]. Under Cd stress, the rate of seed germination decreases and photosynthesis is inhibited when cations are displaced [11, 12]. To cope with Cd stress and decrease the toxicity of Cd, plants have evolved various physical and chemical defense systems [13–16]. For example, plants change their carbohydrate content in response to Cd stress [17, 18].

Sucrose, the primary product of photosynthesis in most plants, is an important source of carbon and energy. It is transported through the plant in the phloem from its source to the sink to be used in physiological processes that contribute to plant growth and development and help plants respond to stress [19, 20]. Metabolism of sucrose helps to regulate abiotic stress in plants, such as that from Cd [21–23]. As signal molecules, sucrose and its metabolism products modulate the responses of plants to stresses, either directly or by interacting with hormone and redox signal molecules; also, as osmotic protectants, they can protect biomolecules and membranes in plants and enhance plant tolerance to abiotic stresses [24, 25]. Many enzymes contribute to sucrose metabolism. Sucrose-phosphate synthase (SPS) catalyzes substances for sucrose synthesis [26]. Invertases (INVs), including acid invertase (AI) (the main INVs) and neutral invertase (NI), hydrolyze sucrose into glucose and fructose [27]. Sucrose synthase (SS) catalyzes a reversible reaction that degrades sucrose in plants [28]. The activities of these enzymes change under abiotic stresses [17, 25]. Apoplast loading is the main pathway by which sucrose enters the phloem in maize [29]. Sucrose transporters (SUTs) and sugars will eventually be exported by transporters (SWEETs) that load and unload sucrose in the phloem and translocate and allocate sugar in plants [30–32]. Some sugar transporter genes have displayed this function under abiotic stresses [33–35].

The effects of Cd stress vary considerably between different crop accessions. Liu et al. [36], from their study of Cd absorption and translocation in six rice cultivars, found that Cd accumulated at different levels in different organs of the rice cultivars, especially in the grains. Further, Guo et al. [16] found that the activities of antioxidant enzymes and hormones differed between two Cd-stressed wheat cultivars, and Wu et al. [37] reported similar differences between two oilseed rape cultivars, also under Cd stress. Maize is a very important crop in China. Studies to date have shown that Cd can affect the metal uptake of maize and modify the composition of the cell walls of maize roots [13, 38]. Using Cd concentrations, the genetic basis of Cd accumulation has been investigated in 269 maize accessions in the Genome-Wide Association Study [39]. Despite these studies, the mechanisms that drive changes in sucrose metabolism in maize varieties with different sensitivities to Cd under different Cd stress levels are still poorly understood. In this study, we selected two varieties with different sensitivities to Cd stress, a low sensitivity variety (Fuyou No. 9, FY9) and a high sensitivity variety (Shenyu No. 33, SY33), from sixteen hybrid maize varieties. Then, we examined how the photosynthesis, biomass, and sucrose metabolism in these two hybrid varieties changed in response to different levels of Cd stress. The main aims of the present study were to evaluate the effects of maize varieties with different Cd sensitivities on photosynthesis and material accumulation, compare and analyze the response of sucrose metabolism to Cd stress, and then examine the role of sucrose metabolism in Cd-stressed maize varieties.

## Materials and methods

### Plant material and Cd treatment

We obtained seeds for the SY33 variety from Shenyang Academy of Agricultural Sciences and for the FY9 variety from Dongya Seed Company, Shenyang, Liaoning Province, China. The

seeds were sterilized in a 1% sodium hypochlorite solution (v/v) for 10 min and rinsed with sterile water three times and then soaked in sterile water at 4˚C for 24 h to adsorb water. The seeds were transferred into petri-plates lined with three wet filter papers and placed in an incubator at 28˚C in the dark to germinate for 3 days. The germinated uniform seeds were transferred into plastic containers and cultivated in a 1/4 Hoagland solution (5 mM $KNO_3$, 2 mM $NH_4H_2PO_4$, 2 mM $MgSO_4$, 3 mM $Ca(NO_3)_2$, 1 μM $ZnSO_4$, 0.8 μM $CuSO_4$, 9 μM $MnCl_2$, 0.5 μM $(NH_4)_6Mo_7O_{24}$, 46 μM $H_3BO_3$, 100 μM FeEDTA, pH 6.0) at 28˚C with a 16/8 h light/dark photoperiod for 7 days (three-leaf stage). Then the seedlings were treated in the 1/4 Hoagland solution with different concentrations of $CdCl_2$ (0, 5, 10, and 20 mg $L^{-1}$) for 18 days (six-leaf stage). The nutrient solution was changed every two days. Tissues were collected and stored in liquid nitrogen after treatment. Each treatment was repeated at least four times with three seedlings in each replicate.

## Photosynthetic parameter determination

The photosynthetic rate was measured using a portable photosynthesis system (Li-6800, Licor, US). We measured the rate at the middle of the uppermost mature leaf at least three times from five seedlings for each treatment.

The chlorophyll was extracted and measured using method of Ma et al. [40]. Approximately 0.2 g of fresh leaf samples were ground to powder using quartz sand and then added to 15 mL acetone (80%) and incubated at 4˚C until the sample turned white. Then the samples were filtered and the volume was made up to 25 mL with 80% acetone. The absorbance was measured at 645 and 663 nm using a UV-T6 spectrophotometer, and the chlorophyll content is expressed in mg $L^{-1}$.

## Determining the biomass, Cd, and sugars

The biomass was determined by measuring the dry weight of shoots and roots. After the treatments, the fresh tissues were dried at 80˚C until the weight was constant, and then cooled in the dryer.

The dry plant tissues were mixed with $HNO_3$ and $HClO_4$ (v/v, 83/17) for 24 h. The mixtures were then digested at 90˚C for 3 h, 150˚C for 5 h, and 180˚C until nearly dry. The digested products were cooled and dissolved in $ddH_2O$ to reach a total volume of 25 mL. The Cd concentrations were measured using an atomic absorption spectrophotometer.

To estimate the sugar, the shoot and root samples were dried at 105˚C for 30 min, and at 70˚C for 24 h. They were then homogenized in 80% ethanol, boiled at 70˚C for 30 min, and centrifuged at 8000 g at 4˚C for 10 min. The total soluble sugars in the supernatants were measured using the method of McCready et al. [41]. The fructose, glucose, and sucrose contents were measured using high performance liquid chromatography (HPLC, Waters 600 HPLC), as described by Sánchez-Linares Luis et al. [42].

## Protein extractions and enzyme assays

The fresh tissues (1 g) were ground to powder using quartz sand to obtain homogenate and then mixed with a 50 mM phosphate buffer (pH 7.5) at 4˚C. The homogenate mixture was centrifuged at 12,000 g for 20 min at 4˚C. The supernatant was divided into two portions, one of which was stored, while the other was analyzed for activities of SPS, SS, and NI, following method of Saher et al. [43]. The precipitate was resuspended in the same extraction buffer with 1 M KCl and agitated continuously for 60 min at 4˚C. After centrifugation at 12,000 g for 20 min, the supernatant was mixed with the stored supernatant solution. The activities of AI were measured on this mixture, following method of Saher et al. [43].

### RNA extraction and RNA analysis

RNA was isolated from different tissues using an RNAprep pure Plant Kit (Qiagen). The total RNA (about 2 μg) was reverse transcribed to obtain synthetic cDNA using a MMLV reverse transcriptase (Promega). Quantitative reverse polymerase chain reaction (qRT-PCR) assays were performed with a real-time PCR detection system (CFX96 Bio-RAD) using SuperReal PreMix Plus Kit (Qiagen). The PCR was initially held at 95°C for 15 min, followed by 40 cycles of 95°C for 10 s, 60°C for 20 s, and 72°C for 30 s. The ΔΔCT method was used to analyze the transcript levels of the relevant genes [44]. The *ZmTubulin1* (ID: Zm00001d013367) gene was used as an internal control for data normalization. The primers of *ZmTubulin1*, *ZmSWEET13a* (ID: *Zm00001d023677*), *ZmSWEET13b* (ID: *Zm00001d023673*), *ZmSWEET13c* (ID: *Zm00001d041067*), *ZmSUT1* (ID: *Zm00001d027854*), and *ZmSUT4* (ID: Zm00001d041192) are listed in Table 1. The experiments were repeated three times.

### Statistical analysis

All data are shown as means and the standard deviation (SD). Two-way analysis of variance (ANOVA) and Duncan's multiple range test were carried out using SPSS version 26. The significance level was $P < 0.05$. Each experiment had four replicates.

## Results

### Changes in biomass and Cd accumulation in two maize varieties

By comparing the controls with the Cd-treated plants, the negative effects of the Cd were obvious. The shoots and roots of both varieties were affected by Cd at different concentrations. The dry weight (DW) of shoots and roots of FY9 were between 12.2%–28.4% and 10.2%–24.2% less than those of the control, while DW of shoots and roots of SY33 were between 18.5%–58.5% and 16.5%–45.5% less than those of the control, respectively. The root and shoot ratios of FY9 and SY33 were between 0.167–0.2 and 0.189–0.249, respectively (Table 2).

We determined the Cd concentrations in shoots and roots, to identify how Cd accumulated and was distributed in both varieties. We found that the Cd accumulation increased in shoots and roots of maize plants as the Cd concentrations increased. More Cd accumulated in the shoots of SY33 than in FY9 (Fig 1A), but less Cd accumulated in roots of SY33 than in FY9 (Fig 1B). Cd concentration was significantly influenced by cultivar, Cd treatment and

**Table 1. Sequence of the primers used in this study.**

| Gene name | Primer mane | Primer sequence (5'-3') |
|---|---|---|
| *ZmTubulin1* | ZmTubulin1-F | GTGTCCTGTCCACCCACTCTCT |
| | ZmTubulin1-R | GGAACTCGTTCACATCAACGTTC |
| *ZmSWEET13a* | ZmSWEET13a-F | GGCTTCCAGTCGGTTCCCTA |
| | ZmSWEET13a-R | AGACGCCGCCATTGAGGA |
| *ZmSWEET13b* | ZmSWEET13b-F | GGTTTCCAGTCGGTTCCC |
| | ZmSWEET13b-R | AGACGCCGCCATTGAGGA |
| *ZmSWEET13c* | ZmSWEET13c-F | GCGACCAAGAAGGGCAGGAT |
| | ZmSWEET13c-R | GGAGAAGCCGACGCAGAT |
| *ZmSUT1* | ZmSUT1-F | TCGGCAAGGGCAACATCC |
| | ZmSUT1-R | TTGGGCAGCAGGAACACG |
| *ZmSUT4* | ZmSUT4-F | GCCATCTGCGTCTACCTTG |
| | ZmSUT4-R | GCGATCCGAGTCCTCCTT |

**Table 2. The effect of different Cd concentrations on the biomass (dry weight) of the two maize varieties.**

| Cultivar | Cd Treatment (mg L$^{-1}$) | Biomass (DW) | | |
|---|---|---|---|---|
| | | Shoot (g) | Root (g) | Root/Shoot |
| FY9 | 0 | 0.764 ± 0.066a | 0.128 ± 0.012a | 0.167 ± 0.0018c |
| | 5 | 0.671 ± 0.07ab | 0.115 ± 0.011a | 0.171 ± 0.015c |
| | 10 | 0.597 ± 0.047bcd | 0.112 ± 0.014a | 0.187 ± 0.012c |
| | 20 | 0.547 ± 0.016cd | 0.110 ± 0.014a | 0.2 ± 0.021bc |
| SY33 | 0 | 0.639 ± 0.185bc | 0.121 ± 0.036a | 0.189 ± 0.014bc |
| | 5 | 0.521 ± 0.032d | 0.101 ± 0.012ab | 0.195 ± 0.035bc |
| | 10 | 0.335 ± 0.042e | 0.079 ± 0.005bc | 0.237 ± 0.039ab |
| | 20 | 0.265 ± 0.021e | 0.066 ± 0.011c | 0.249 ± 0.036a |
| F (Cultivars) | | 75.576** | 16.885* | 13.212** |
| F (Cd) | | 31.591** | 8.088* | 5.07* |
| F (Cultivars × Cd) | | 2.766 | 2.138 | 0.614 |

Data shown are the means of four independent replicates. Values are the means ± SD (n = 4). Different letters indicate a significant difference from each other according to two-way ANOVA followed by Duncan multiple comparison P < 0.05. F-test:

*P < 0.05

**P < 0.01.

cultivar × Cd in shoot and root (Fig 1). These results show that Cd accumulated in the roots and shoots of the maize seedlings, but to different levels.

## Effect of Cd on photosynthesis of two maize varieties

Photosynthesis contributes to biomass accumulation. We detected some photosynthetic parameters in both varieties at different concentrations of Cd to examine how photosynthesis changed under Cd stress. Relative to the control, the photosynthesis rates gradually decreased as the Cd concentration increased and reduced to 55% and 42% in FY9 and SY33 under 20 mg L$^{-1}$ Cd stress, respectively (Fig 2A). The chlorophyll a and b contents of FY9 were 56.9% and 59.3% lower in FY9, and 58.7% and 62.7% lower in SY33, respectively, under high concentrations of Cd (Fig 2B and 2C). The chlorophyll contents and photosynthesis rates of FY9 were always higher than those of SY33 for the same Cd concentration. These data suggest that Cd had a negative influence on photosynthesis in the maize seedlings and had more effect on Cd-sensitive variety than Cd-tolerant variety.

## Effect of Cd on sugar contents of two maize varieties

The sugar contents were examined under different concentrations of Cd to investigate how Cd affected the sucrose metabolism. The total soluble sugars were examined in shoots and roots of FY9 and SY33. The soluble sugar content in shoots of SY33 and FY9 increased as the Cd concentration increased, and increased by 66.7% and 229% in FY9 and SY33 under 20 mg L$^{-1}$ Cd stress, respectively (Fig 3A). The soluble sugar contents also increased in roots of both varieties, but there were no obvious differences between the two varieties for the same Cd concentration (Fig 3B). The sucrose and glucose contents changed slightly in shoots and roots of FY9 as Cd concentration increased, with greater increases in FY9 than in SY33 (Fig 3C–3F). There was no obvious change in the fructose contents in roots and shoots of FY9 or roots of SY33. The fructose contents in shoots of SY33 were 155% higher than in the control under 20 mg L$^{-1}$ Cd stress, and were significantly higher than those in FY9 after the Cd treatments (Fig 3G and 3H). In addition, soluble sugar, sucrose and fructose contents were significantly influenced by

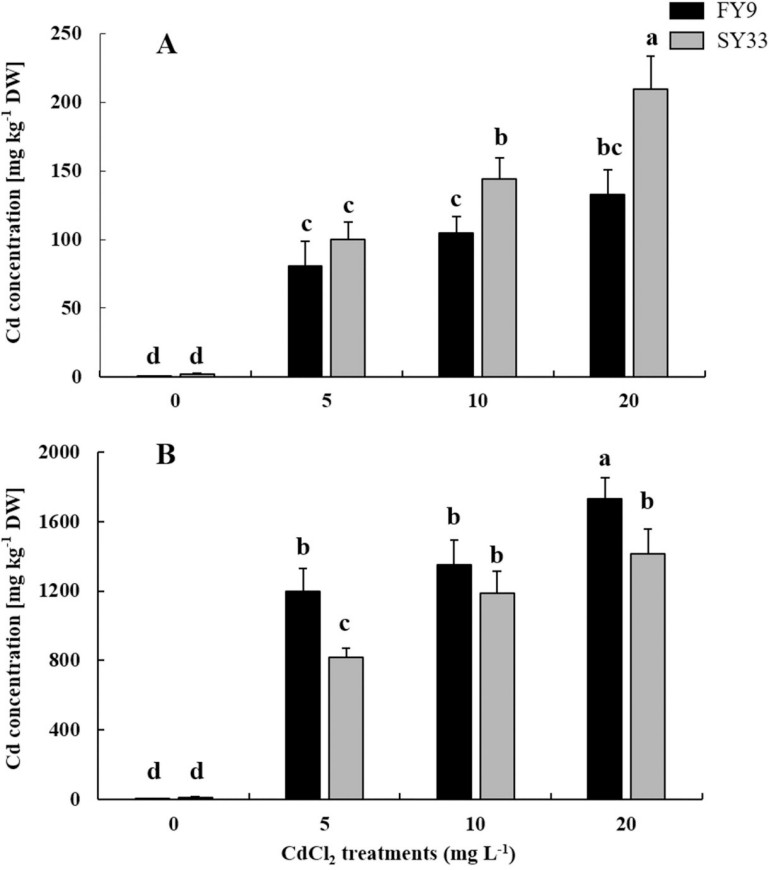

**Fig 1.** Cd concentrations in the shoots (A) and roots (B) of two maize varieties. Data are presented as the treatment mean ± SD (n = 4). Different letters indicate significant differences at the P < 0.05 level (Duncan's test). F-test: Cd concentration: Shoot: F(Cultivars) = 30.54**; F(Cd) = 134.57**; F(Cultivars × Cd) = 6.84**. Root: F(Cultivars) = 48.43**; F(Cd) = 489.93**; F(Cultivars × Cd) = 7.92**. *P < 0.05; **P < 0.01.

cultivar, Cd treatment and cultivar × Cd in shoot, and glucose and sucrose contents were also significantly influenced in root. These results show that the soluble sugars changed as Cd concentrations changed and fructose accumulated in the shoots of Cd-sensitive variety.

## Effect of Cd on activities of sucrose metabolism enzymes in two maize varieties

The activities of SPS, Sus, and INV were examined to gain insights into how the sucrose metabolism processes changed in FY9 and SY33 as Cd concentration increased. The activities of SPS in shoots and roots gradually decreased as Cd concentration increased (Fig 4A and 4B). AI, the main invertase in plants, is found in cell walls and vacuoles, while NI exists in cytoplasm. The AI activity increased considerably in shoots of FY9 and SY33 as Cd concentration increased, and was higher in SY33 than in FY9 (Fig 4C), but did not change in roots of the two varieties (Fig 4D). Sus catalyzes the reversible conversion of sucrose to glucose and fructose. The SS activity in the sucrose hydrolysis direction (SS-H) was analyzed in shoots and roots of both varieties as Cd concentrations increased. There was no obvious change in the activity in shoots of FY9 and roots of FY9 and SY33, but it was higher in shoots of SY33 at high Cd concentrations than in the control (Fig 4E and 4F). The activity of SS in the synthesis direction (SS-S) did not change in shoots of FY9, but it was considerably lower in shoots and roots of

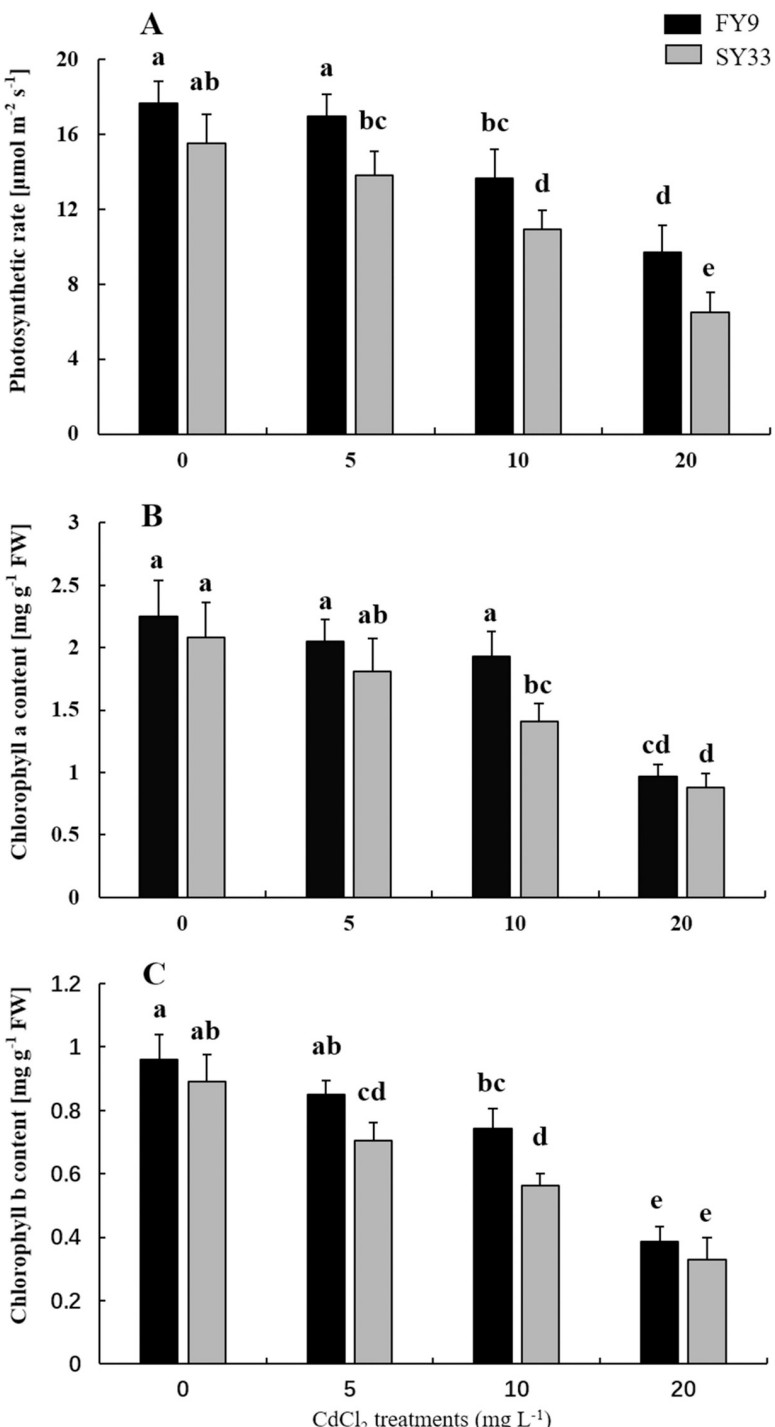

**Fig 2.** Effects of different Cd concentrations on the (A) photosynthetic rate, chlorophyll (B) a and (C) b contents of the two maize varieties. Data are presented as the treatment means ± SD (n = 4). Different letters indicate a significant difference when $P < 0.05$ (Duncan's test). F-test: Photosynthetic rate: F(Cultivars) = 55.54**; F(Cd) = 102.15**; F (Cultivars × Cd) = 0.41. Chlorophyll a: F(Cultivars) = 17.11**; F(Cd) = 80.63**; F(Cultivars × Cd) = 2.36. Chlorophyll b: F(Cultivars) = 39.60**; F(Cd) = 181.13**; F(Cultivars × Cd) = 2.63. *P < 0.05; **P < 0.01.

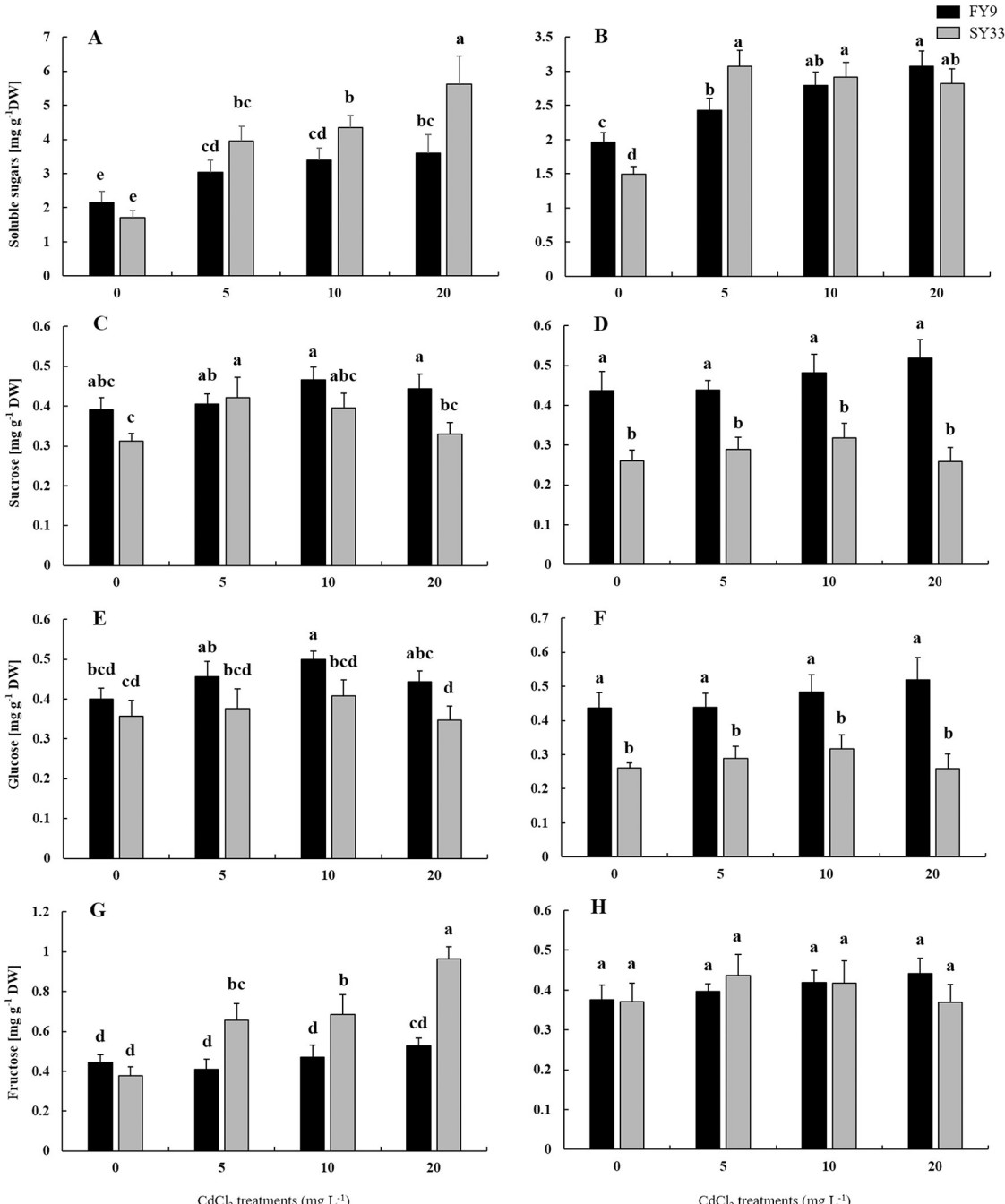

**Fig 3.** Soluble sugar contents in (A) shoots and (B) roots, sucrose contents in (C) shoots and (D) roots, glucose contents in (E) shoots and (F) roots, and fructose contents in (G) shoots and (H) roots of two maize varieties with different concentrations of Cd. Data are presented as the treatment means ± SD (n = 4). Different letters indicate a significant difference when $P < 0.05$ (Duncan's test). F-test: Soluble sugar: Shoot: F(Cultivars) = 41.95**; F(Cd) = 72.32**; F(Cultivars × Cd) = 14.59**. Root: F(Cultivars) = 0.11; F(Cd) = 104.95**; F(Cultivars × Cd) = 19.10**. Sucrose: Shoot: F(Cultivars) = 39.99**; F(Cd) = 12.32**; F(Cultivars × Cd) = 7.92**. Root: F(Cultivars) = 298.49**; F(Cd) = 4.58*; F(Cultivars × Cd) = 5.19*. Glucose: Shoot: F(Cultivars) = 56.02**; F(Cd) = 9.93**; F(Cultivars × Cd) = 1.31. Root: F(Cultivars) = 219.45**; F(Cd) = 3.39*; F(Cultivars × Cd) = 3.84*. Fructose: Shoot: F(Cultivars) = 131.27**; F(Cd) = 58.23**; F(Cultivars × Cd) = 32.67**. Root: F(Cultivars) = 0.59; F(Cd) = 2.88; F(Cultivars × Cd) = 3.52*. *P < 0.05; **P < 0.01.

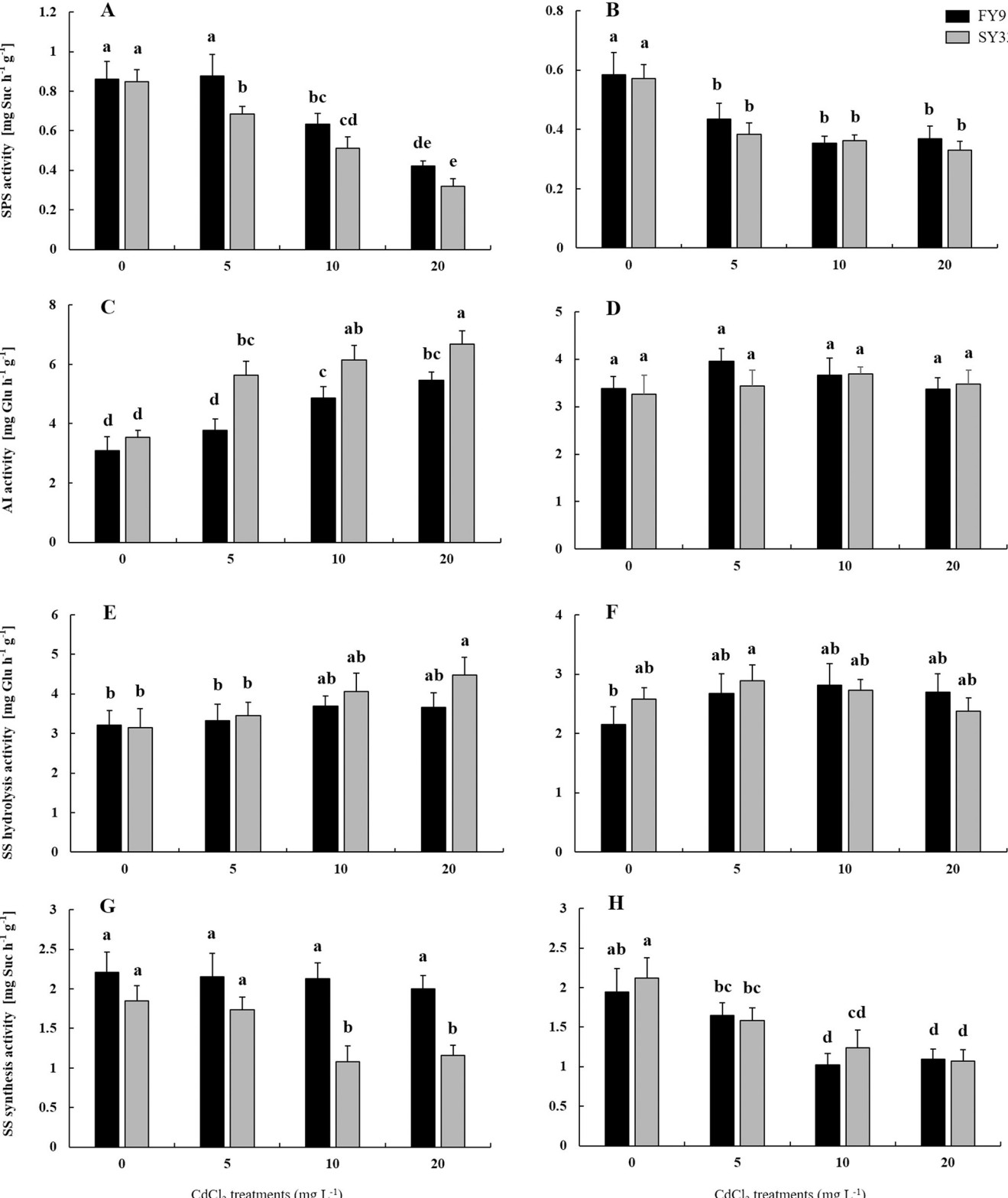

**Fig 4.** Sucrose phosphate synthase (SPS) activity in (A) shoots and (B) roots, acid invertase (AI) activity in (C) shoots and (D) roots, sucrose synthase (SS) hydrolysis activity in (E) shoots and (F) roots, and SS synthesis activity in (G) shoots and (H) roots of two maize varieties at different concentrations of Cd. Data are presented as treatments mean ± SD (n = 4). Different letters indicate a significant difference when $P < 0.05$ (Duncan's test). F-test: SPS activity: Shoot: F(Cultivars) = 33.28**; F(Cd) = 137.86**; F(Cultivars × Cd) = 3.95*. Root: F(Cultivars) = 3.54; F(Cd) = 69.3**; F(Cultivars × Cd) = 1.03. AI activity:

Shoot: F(Cultivars) = 105.07\*\*; F(Cd) = 105.5\*\*; F(Cultivars × Cd) = 6.1\*\*. Root: F(Cultivars) = 2.3; F(Cd) = 4.61\*; F(Cultivars × Cd) = 2.59. SS-H activity: Shoot: F(Cultivars) = 7.51\*; F(Cd) = 12.93\*\*; F(Cultivars × Cd) = 2.78. Root: F(Cultivars) = 0.47; F(Cd) = 6.19\*\*; F(Cultivars × Cd) = 4.09\*. SS-S activity: Shoot: F(Cultivars) = 126.42\*\*; F(Cd) = 15.18\*\*; F(Cultivars × Cd) = 7.84\*\*. Root: F(Cultivars) = 1.89; F(Cd) = 62.31\*\*; F(Cultivars × Cd) = 1.04. \*$P < 0.05$; \*\*$P < 0.01$.

SY33 and in roots of FY9 at high concentrations of Cd than in the control (Fig 4G and 4H). SPS, AI and SS-S activities were significantly influenced by cultivar, Cd treatment and cultivar × Cd in shoot. These data show that sucrose synthesis decreased, and hydrolysis increased, in both varieties after Cd was added.

## Effect of Cd on transcript levels of sugar transporters in two maize varieties

We examined the transcript levels of the main sugar transporters under Cd stresses to examine how Cd affected sugar translocation and allocation. *ZmSWEET13a*, *ZmSWEET13b*, and *ZmSWEET13c* are the main members of the SWEET family in maize [48]. The transcript level of *ZmSWEET13b* increased considerably in shoots of FY9 as the concentrations of Cd increased, and the transcript levels of *ZmSWEET13a* and *ZmSWEET13c* increased in shoots of SY33 as the Cd concentration increased (Fig 5A and 5B). The transcript levels of *ZmSWEET13b* were higher in the roots of FY9, but the transcript levels of all three sweet proteins were lower in roots of SY33 than in the control (Fig 5C and 5D).

*ZmSUT1* and *ZmSUT4* are the main sucrose transporters in maize [31]. The expression of *ZmSUT1* was higher in shoots under Cd treatment, while it was lower than control in roots of FY9 under 20 mg L$^{-1}$ Cd-treatment. The expressions of *ZmSUT1* and *ZmSUT4* were higher in

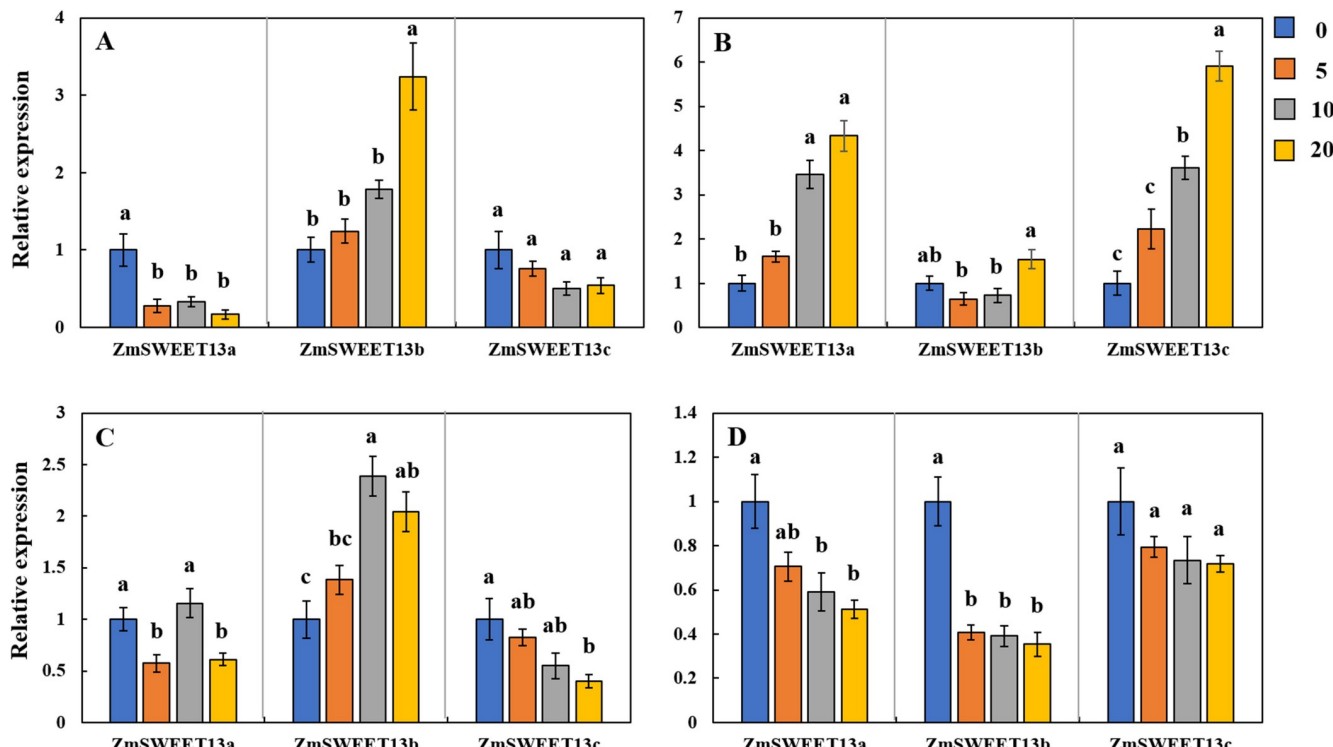

**Fig 5.** Expression profiles of three SWEET13 genes in (A) the shoots of FY9, (B) in the shoots of SY33, (C) in the roots of FY9, and (D) in the roots of SY33 under different Cd concentration treatments (blue: 0 mg L$^{-1}$; orange: 5 mg L$^{-1}$; gray: 10 mg L$^{-1}$; yellow: 20 mg L$^{-1}$). Data are presented as treatment means ± SD (n = 3). Different letters indicate a significant difference when $P < 0.05$ (Duncan's test).

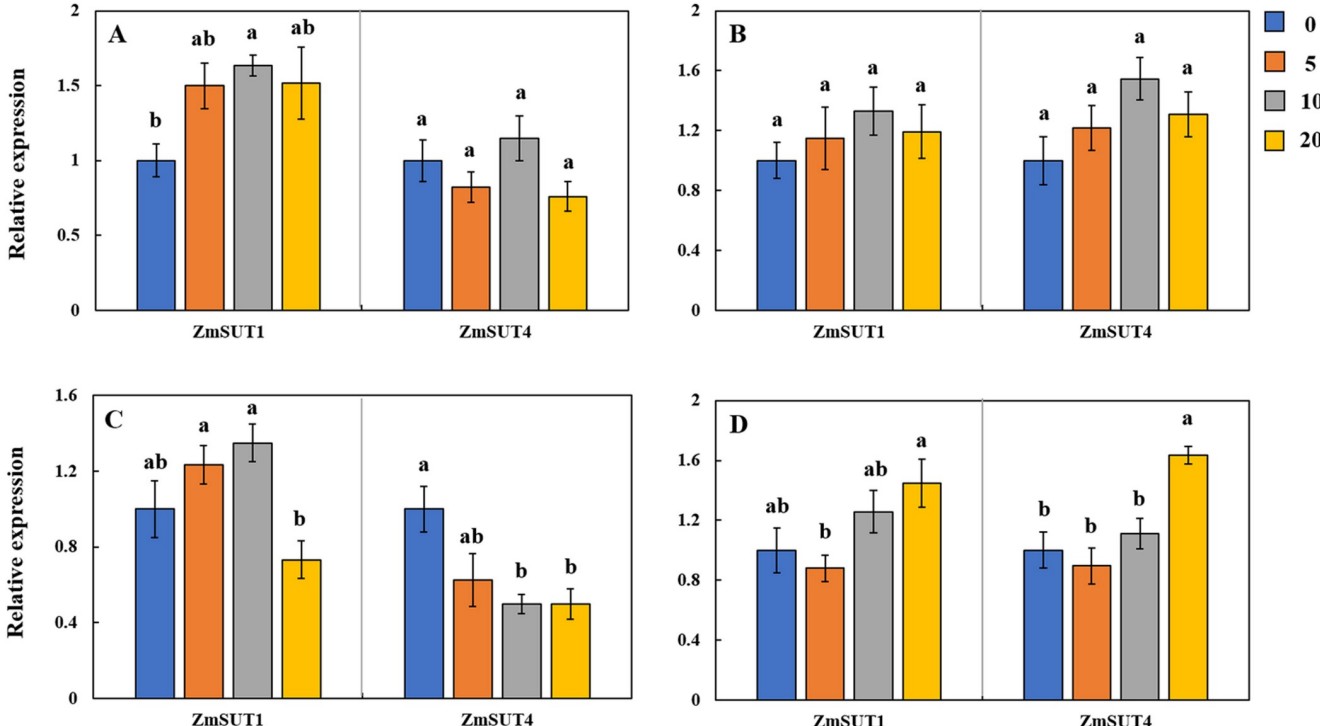

**Fig 6.** Expression profiles of two SUT genes in (A) the shoots of FY9, (B) the shoots of SY33, (C) the roots of FY9, and (D) the roots of SY33 at different Cd concentration treatments (blue: 0 mg $L^{-1}$; orange: 5 mg $L^{-1}$; gray: 10 mg $L^{-1}$; yellow: 20 mg $L^{-1}$). Data are presented as the treatment means ± SD (n = 3). Different letters indicate significant difference when $P < 0.05$ (Duncan's test).

shoots and roots of Cd-treated SY33, than in the control (Fig 6A–6D). These results imply that the transcript levels of the sugar transporters that regulated sugar transport were affected by Cd additions.

## Discussion

With no physiological function, Cd is toxic to plants. Cd competes with essential nutrient ions and limits their absorption, which in turn inhibits the activities of key metabolic enzymes. Therefore, it affects the metabolic system, inhibits the growth and development of plants, and causes the biomass to decrease [39]. The results of this study show that the biomass of two maize varieties (FY9 and SY33) decreased as Cd was added, with the tendency increasingly obvious as Cd concentrations increased, as also reported by Xu et al. [45] and Guo et al. [16]. Cd had more effect on the biomass of shoots and roots of the Cd-sensitive cultivar SY33 than on those of the Cd-tolerant cultivar FY9. Also, Cd had more impact on the biomass of shoots of both varieties than on the roots. The results were similar to those reported for *Brassica rapa* and maize [46, 47], and indicate that FY9 was more tolerant of, and SY33 was more sensitive to, Cd stress, and that Cd caused more damage to shoots than roots in both varieties.

Cd is first absorbed by the plant roots, and then is transported to the above-ground parts of plants. To prevent the plant from Cd toxicity and to protect photosynthetic tissues, most Cd is stored in the root cell walls and vacuoles [48–50]. We found, however, that most of Cd tended to accumulate in roots of both varieties, and only a little was transported to the maize shoots. Also, the chlorophyll a and b contents reduced and photosynthesis rates decreased, which means that photosynthesis was inhibited under Cd stress. Meanwhile, we found that Cd

accumulation varied in the two varieties. For example, more Cd accumulated in shoots of SY33 than in those of FY9 when Cd concentrations were high, and so the photosynthesis in SY33 was affected more than that of FY9. However, when Cd concentrations in shoots of the two varieties were similar, the photosynthetic rates and chlorophyll contents were similar, as occurred for Cd treatments of 10 mg/L in FY9 and 5 mg/L in SY33 (Fig 2). More Cd was absorbed and immobilized in roots of FY9 than in roots of SY33. This means that less Cd was translocated to the shoots of FY9, with less effect on photosynthesis in FY9 than SY33, leading to more photosynthetic production synthesized in FY9 than in SY33.

Many studies have shown that, when plants are subjected to Cd stress, the activities of the main enzymes of the sucrose metabolic pathway are affected [12, 51, 52]. In our study, the SPS activities in both varieties were lower under Cd additions than in the control, as also reported by Shah and Dubey [52] from their study of rice. Under Cd stress, the SS-S activity was lower in SY33, but showed no obvious change in shoots of FY9, relative to the control, perhaps because Cd at low concentrations had little impact on SS-S activity in shoots of FY9. The SPS activity was more sensitive than SS-S activity in shoots of FY9 at the same Cd concentration. The AI and SS-H activities were higher in shoots than in the control, and were higher in SY33 than that in FY9 as Cd concentration increased. The sugar contents are regulated by the enzymes of the sucrose metabolic. The increase in the sugar content could maintain the osmotic balance in cells [24, 25]. For example, Verma and Dubey [17] and Sfaxi-Bousbih et al. [12] reported that sugar content increased in bean cotyledons and rice under Cd stress, respectively. Cd toxicity induces a defense response, such as oxidative stress and osmotic stress. In our study, the total soluble sugar and fructose contents, osmotic protectants in shoots of both varieties, increased as the Cd concentration increased, with a greater increase in SY33 than FY9 (Fig 3A and 3G), perhaps because Cd stress meant that less water was adsorbed and transported to the shoot, which then triggered ABA as a signal of dehydration stress. Therefore, sucrose metabolism was regulated by the sucrose metabolic enzymes to produce more soluble sugars to maintain osmotic balance in plants. In addition, Cd may cause osmotic stress and induce the expression of dehydration relative genes in barley [53]. Meanwhile, water transport was limited from the roots to the shoots. Therefore, Cd had more effect on the shoots than the roots under the same Cd concentration.

Different varieties have different tolerances to Cd stress. Comparison of the two varieties showed that SY33 was more sensitive to Cd and accumulated more sugars in its shoots than FY9. This suggests that SY33 suffered more damage and needed more soluble sugar to maintain osmotic balance of its cells when stressed by Cd. Cd impairs the ability of a plant to transport sugar from its source to where it is needed for growth [54]. The total soluble sugar contents showed a clear increase when the Cd concentrations were low in the roots of the Cd-sensitive maize, SY33. However, as Cd concentrations increased, total soluble sugar contents gradually decreased. This suggests that, in SY33, the sugar was transported from the shoots to the roots under low Cd concentrations, but was obstructed at high Cd concentrations. The sugar mobilization was not inhibited in shoots of FY9, the Cd-insensitive maize, and the soluble sugar contents gradually increased as Cd concentrations increased. So, in this study, sugar transport from the source (leaves) to the sink (roots) was promoted when the Cd concentration was low, but was inhibited when the Cd concentration was high in Cd-sensitive variety. This transport was not inhibited in Cd-insensitive variety, which suggests that Cd levels in the roots of the insensitive variety did not impair the sugar transportation. Sugar transporters are also involved in translocating and allocating sugar in plants [30–32]. In our experiment, we examined the transcript levels of three main SWEET genes at different Cd concentrations. The transcript levels of *ZmSWEET13b* increased significantly, but those of *ZmSWEET13a* and *ZmSWEET13c* decreased, as Cd concentration increased (Fig 5A and 5C). The transcript levels

of *ZmSWEET13a* and *ZmSWEET13c* were higher in the shoots of Cd-treated SY33 than in the control (Fig 5B). These results suggest that *ZmSWEET13b* may be the main transporter in the Cd-treated FY9, while *ZmSWEET13a* and *ZmSWEET13c* may be the main SWEET genes in the Cd-treated SY33. The transcript levels of *SUT1* and *SUT4* were slightly higher in the Cd-treated seedlings than in the control. The SUT genes may maintain the fundamental sugar transport, while the functions of the SWEET genes may change in response to the Cd levels.

In conclusion, the photosynthesis was impaired, and the production of photosynthesis decreased, which means that fewer substrates were involved in sucrose metabolism under Cd stress. Meanwhile, the activities of sucrose metabolism enzyme change as the Cd concentration increased, causing increases in the soluble sugars produced, transported, and stored in the shoots and roots. That means more carbohydrates, which came from photosynthesis and were regulated by sucrose metabolic enzymes, were used to resist Cd stress rather than to support growth. Especially in Cd-sensitive variety, SY33 needed to accumulate more soluble sugars to maintain the osmotic balance in cells than the Cd-insensitive variety, FY9, as Cd concentration increased. Hence, sucrose metabolism is more active in Cd-sensitive variety than in Cd-insensitive variety, and so may be a secondary response that helps the maize survive in Cd-stressed conditions.

## Supporting information

**S1 Table. The changes of Cd concentration, photosynthetic rate, chlorophyll contents, sugar contents, enzyme activities under different concentration Cd treatments.** (XLSX)

## Author Contributions

**Conceptualization:** Cong Li, Yu Liu, Yanshu Zhu, Jinjuan Fan.

**Formal analysis:** Yu Liu, Jing Tian.

**Investigation:** Yu Liu, Jing Tian.

**Resources:** Cong Li, Jinjuan Fan.

**Supervision:** Jinjuan Fan.

**Writing – original draft:** Cong Li.

**Writing – review & editing:** Yanshu Zhu, Jinjuan Fan.

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
