## [Decision Letter · Decision Letter 0]

3 Aug 2020

PONE-D-20-20550

Changes in sucrose metabolism in maize varieties with different Cd sensitivities under Cd stress

PLOS ONE

Dear Dr. Fan,

Thank you for submitting your manuscript to PLOS ONE. After careful consideration, we feel that it has merit but does not fully meet PLOS ONE’s publication criteria as it currently stands. Therefore, we invite you to submit a revised version of the manuscript that addresses the points raised during the review process.

Although the theme of the article is interesting, yet there are numerous flaws in different sections of the draft including introduction, methods, results and discussion. A detail revision according the the comments of both reviewers in inevitable along with the following suggestions.

Avoid abbreviation in the title. Introduction section contains various unnecessary statements. Remain focused on the topic. Begin with the broadest scope and get progressively narrower, leading steadily to the statement of objectives. Be clear regarding objectives.

Material and Method section: Several points need to be clarified. The information regarding statistical analysis and experimental design should be made clear.

In results, better to add the numeric description of results (% variations).

Discussion should be merely based on the observed findings. Not just a review of literature. Answer the question posed in introduction and correlate your finding with the existing knowledge.

Language needs substantial improvement. There are several grammatical and typo mistakes throughout the manuscript.

We look forward to receiving your revised manuscript.

Kind regards,

Saddam Hussain

Academic Editor

PLOS ONE

Journal Requirements:

"This work was financially supported by the scientific research projects of Liaoning

Provincial Department of Education, China (No. LSNZD201608 and

LSNYB201607)."

Reviewers' comments:

Reviewer's Responses to Questions

**Comments to the Author**

1. Is the manuscript technically sound, and do the data support the conclusions?

Reviewer #1: Partly

Reviewer #2: Yes

2. Has the statistical analysis been performed appropriately and rigorously? 

Reviewer #1: I Don't Know

Reviewer #2: Yes

3. Have the authors made all data underlying the findings in their manuscript fully available?

Reviewer #1: Yes

Reviewer #2: Yes

4. Is the manuscript presented in an intelligible fashion and written in standard English?

Reviewer #1: No

Reviewer #2: Yes

5. Review Comments to the Author

Reviewer #1: After peer reviewing, I found following shortfalls in the overall structure of MS that must be considered according to the standards of journal:

The Cd concentration @ 20 mg/L used in this study is too high but could be compromised because authors selected it as one of the treatments to study Cd toxicity effects on maize varieties. However, the surprising thing is that this higher toxicity of Cd did not significantly affect the biomass of root or root shoot ratio in FY9 cultivar relative to control Table 2. Please explain the possibilities for these results?

Similarly most of the investigated parameters e.g. sucrose, fructose, glucose and similarly enzymatic activities were not significantly affected with higher Cd treatments relative to control in root tissues than shoot. Please could authors prove these results with some references that Cd toxicity has negligible effects on root relative to shoots in crop plants?

Introduction doesn’t end with specific objectives of the study?

Hoagland solution composition is not mentioned.

Statistical analysis is not explained.

Results are more general and based on well-established reports e.g. Pn decreased, sugar contents increased or Cd concentrations also increased with the increase of Cd treatments.

Discussion should explain the possible reasons for the determined parameters (increase or decrease) but this part is not properly elucidated, so this part should be rewritten with probable reasoning.

MS needs significant improvement regarding English language as there are lots of typographic or language mistakes that are difficult to highlight in present form.

Reviewer #2: In the last past paragraph of conclusion, need to cite references for the following statements:

" the soluble sugars that come from sucrose metabolism products have important roles as osmotic protectants and signal molecules that defend against Cd stress,"

What does this mean? --"sucrose metabolism may be a passive response " Why is passive? this is in need some explanation.

It would be more helpful to put in Cd-tolerant and Cd-sensitive cultivars rather than FY9 and SY33, respectively, in the Abstract, results and discussion.

In the methods and materials, there should be statistical analysis, and info on replicates.

6. PLOS authors have the option to publish the peer review history of their article (what does this mean?). If published, this will include your full peer review and any attached files.

Reviewer #1: **Yes: **Muhammad Imran

Reviewer #2: **Yes: **David W.M. Leung

---

## [Author Response · Author response to Decision Letter 0]

19 Sep 2020

Dear Editor

We would like to thank you for your letter and for the reviewers’ comments and constructive suggestions about our manuscript entitled “Changes in the sucrose metabolism of maize varieties with different Cd sensitivities under Cd stress” (ID: PONE-D-20-20550). 

We found that your comments were useful and helped us improve our manuscript. We have carefully considered your comments and have revised the manuscript to incorporate your suggestions. The revised parts are marked in red in the paper. We have compiled the main corrections in the paper and the responses to the reviewers’ comments below. We hope that the corrections will meet your approval. Again, we would like to express our appreciation for your comments and we look forward to hearing from you in due course. 

Responses to the reviewers’ comments:

Reviewer #1: 

1. The Cd concentration 20 mg/L used in this study is too high but could be compromised because authors selected it as one of the treatments to study Cd toxicity effects on maize varieties. However, the surprising thing is that this higher toxicity of Cd did not significantly affect the biomass of root or root shoot ratio in FY9 cultivar relative to control Table 2. Please explain the possibilities for these results?

Response: We think this is an excellent question. It is very important to select the appropriate concentration of Cd to treat samples in the experiment. In this study, we treated the seedlings with different concentrations of CdCl2, namely 5, 10, and 20 mg/L. Because the molecular weight of CdCl2 is 183.32, the molar concentration of 20 mg/L CdCl2 is about 100 μM/L. Because the treatment times and materials varied, the Cd concentrations in the treatment were different, and the Cd concentrations ranged from 1 μM/L to 500 μM/L. Other researchers have treated crops with Cd at high concentrations, for example, Verma et al. (2001) treated rice with 500 μM/L CdCl2 for 20 days and Wen et al. (2018) treated maize with 200 mg/L CdCl2•2.5H2O for 3 days. Therefore, we considered that a concentration of 20 mg/L CdCl2 was appropriate for treating the maize for 18 days. 

In previous studies, we examined how 16 maize varieties that are widely grown in the northeast of China responded to Cd stress according to the changes of phenotype. We found that FY9 and SY33 represented low and high sensitivities varieties, respectively. The growth of FY9 was not significantly inhibited when Cd was added, but the biomass of the shoots and roots decreased by 28% and 15%, respectively. Liu et al. (2007) found that different accessions had different tolerances to Cd street. Also, different plant organs responded differently to Cd stress. For example, Kiran et al. (2020) found that the dry weight of Brassica rapa ssp. chinensis L. (pak choi) leaves decreased by more than that of the roots. Therefore, the difference in the genotype meant that FY9 had low sensitivity to Cd stress, and, in this variety, more photosynthetic products were used to accumulate material rather than resist Cd stress when Cd was added, especially to roots. 

2. Similarly most of the investigated parameters e.g. sucrose, fructose, glucose and similarly enzymatic activities were not significantly affected with higher Cd treatments relative to control in root tissues than shoot. Please could authors prove these results with some references that Cd toxicity has negligible effects on root relative to shoots in crop plants? 

Response: We have not been able to find any references that directly prove that Cd toxicity has less effect on roots than shoots at the physiological level in the crops. We did however find that our results for changes in biomass were similar to those of other studies, and that the reductions in biomass were more pronounced in shoots of maize and Brassica rapa than in the roots under Cd stress (Abdelgawad et al., 2020, Kiran et al., 2020). Different organs had different sensitivities to Cd stress in plants. Stress from heavy metals limits the nutrient uptake and affects the water transport from the roots to the shoots and upsets the water balance in plants (Schat et al., 1997). The shoots were more sensitive to water stress than the root so the sugar contents and enzymatic activities of sucrose metabolism in the shoots were more significantly affected than the roots, as they tried maintain the osmotic balance. 

3. Introduction doesn’t end with specific objectives of the study?

Response: Line 88-92. We have added the objectives of the study in the introduction of the unmarked version.

4. Hoagland solution composition is not mentioned.

Response: Line 103-105. We have added information about the Hoagland solution composition in the materials and methods of the unmarked version.

5. Statistical analysis is not explained.

Response: We apologize for this omission. Line 167-171. We have added information about the statistical analysis to the materials and methods of the unmarked version.

6. Results are more general and based on well-established reports e.g. Pn decreased, sugar contents increased or Cd concentrations also increased with the increase of Cd treatments. 

Response: Yes, the results of our study, such as the decreased Pn, increased sugar contents, and changes in enzyme activities as the Cd concentrations in the two varieties increased, are similar to those published elsewhere. However, we focused on the effects of Cd on sucrose metabolism and compared how the sucrose metabolism varied between maize varieties with low and high Cd sensitivities. Although the trends of sucrose metabolism were similar between two varieties, we found that there were more soluble sugars in Cd highly sensitive variety (SY33) than in FY3 and that the biomass was higher in low sensitivity variety (FY9) than in SY33. This shows that photosynthetic products are used to support maize growth in the low sensitivity varieties under Cd stress, but are used to defend Cd stress and maintain seedlings in highly sensitive varieties.

7. Discussion should explain the possible reasons for the determined parameters (increase or decrease) but this part is not properly elucidated, so this part should be rewritten with probable reasoning.

Response: Thank you for your suggestion. We have rewritten the discussion and explained the reasons for the changes in the parameters.

8. MS needs significant improvement regarding English language as there are lots of typographic or language mistakes that are difficult to highlight in present form.

Response: Thank you for your comment. We have modified and improved the English language and expression in the manuscript.

Reviewer #2:

1. " the soluble sugars that come from sucrose metabolism products have important roles as osmotic protectants and signal molecules that defend against Cd stress,"

What does this mean? --"sucrose metabolism may be a passive response " Why is passive? this is in need some explanation.

Response: We apologize for the ambiguity caused by the word “passive”. We have changed “passive” to “secondary” in lines 31 and 352, respectively. Plants have evolved many defense systems to cope with Cd stress, such as antioxidant systems and osmotic regulation systems. Antioxidant systems produce antioxidants and enzymes to remove the excess ROS induced by Cd stress. While the products of sucrose metabolism are mainly used for plant growth. When plants are subjected to severe osmotic stress, sucrose metabolites are used to maintain the osmotic balance in plants. Cd stress is more serious in highly sensitive maize varieties. That might mean that their defense systems are weak, such that they need more more sucrose metabolites to help plants survive under Cd stress. The situation is reversed in low sensitivity varieties of maize.

2. It would be more helpful to put in Cd-tolerant and Cd-sensitive cultivars rather than FY9 and SY33, respectively, in the Abstract, results and discussion.

Response: Thank you for your suggestion. We would like to explain this further. We used “FY9” and “SY33” to represent varieties with low and high sensitivities to Cd stress throughout the manuscript, respectively. Although the words “Cd-tolerant cultivars and Cd-sensitive cultivars” show more clearly the varieties response to Cd stress, they are very long and perhaps too long to use frequently throughout the paper. Hence, we use “Cd-tolerant and Cd-sensitive cultivars” instead of “FY9 and SY33” in the concluding sentence, and in lines 33, 199-200, 217.

3. In the methods and materials, there should be statistical analysis, and info on replicates.

Response: Thank you for your suggestion. We have added information about statistical analysis and the replicates to the materials and methods. 

Editor’s suggestion

1. Avoid abbreviation in the title. Introduction section contains various unnecessary statements. Remain focused on the topic. Begin with the broadest scope and get progressively narrower, leading steadily to the statement of objectives. Be clear regarding objectives.

Response: Thank you for your suggestion. In the title, we have changed Cd to Cadmium. We have added the aims of the study to the introduction of the unmarked version in line 88-92. 

2. Material and Methods section: Several points need to be clarified. The information regarding statistical analysis and experimental design should be made clear.

Response: We have added information about the statistical analysis and Hoagland solution composition to the Material and Methods section in lines 167–171 and 103–105, respectively.

3. In results, better to add the numeric description of results (% variations).

Response: Thank you for your suggestion. We have added the numerical description of the results to the results section in lines 193–196, 207, and 214.

4. Discussion should be merely based on the observed findings. Not just a review of literature. Answer the question posed in introduction and correlate your finding with the existing knowledge.

Response: Thank you for your suggestion. We have rewritten the discussion and explained the reasons for the changes in the parameters. We have also answered the question in the introduction.

5. Language needs substantial improvement. There are several grammatical and typo mistakes throughout the manuscript.

Response: Thank you for your comment. We have modified and improved the English language of the manuscript.

Schat, H., Sharma, S.S., Vooijs, R., 1997. Heavy metal-induced accumulation of free proline in a metal-tolerant and a nontolerant ecotype of Silene vulgaris, 101, 477e482.

---

## [Decision Letter · Decision Letter 1]

7 Oct 2020

PONE-D-20-20550R1

Changes in sucrose metabolism in maize varieties with different cadmium sensitivities under cadmium stress

PLOS ONE

Dear Dr. Fan,

Thank you for submitting your manuscript to PLOS ONE. After careful consideration, we feel that it has merit but does not fully meet PLOS ONE’s publication criteria as it currently stands. Therefore, we invite you to submit a revised version of the manuscript that addresses the points raised during the review process.

ACADEMIC EDITOR: The manuscript has been adequately revised following the recommendations of the reviewers. Nevertheless, there are still few points (as mentioned by the reviewers) which need to be addressed in the manuscript before it can be accepted for publication.

We look forward to receiving your revised manuscript.

Kind regards,

Saddam Hussain

Academic Editor

PLOS ONE

Reviewers' comments:

Reviewer's Responses to Questions

**Comments to the Author**

1. If the authors have adequately addressed your comments raised in a previous round of review and you feel that this manuscript is now acceptable for publication, you may indicate that here to bypass the “Comments to the Author” section, enter your conflict of interest statement in the “Confidential to Editor” section, and submit your "Accept" recommendation.

Reviewer #1: (No Response)

Reviewer #2: All comments have been addressed

Reviewer #3: (No Response)

2. Is the manuscript technically sound, and do the data support the conclusions?

Reviewer #1: Yes

Reviewer #2: (No Response)

Reviewer #3: Partly

3. Has the statistical analysis been performed appropriately and rigorously? 

Reviewer #1: Yes

Reviewer #2: (No Response)

Reviewer #3: Yes

4. Have the authors made all data underlying the findings in their manuscript fully available?

Reviewer #1: Yes

Reviewer #2: (No Response)

Reviewer #3: Yes

5. Is the manuscript presented in an intelligible fashion and written in standard English?

Reviewer #1: No

Reviewer #2: (No Response)

Reviewer #3: No

6. Review Comments to the Author

Reviewer #1: It is appreciated that authors have tried their best to improve the manuscript but still following are some concerns that must be addressed:

There are no line numbers in the present form of ms and so authors should use line number option as it is difficult to highlight the mistakes at specific points

‘The seeds were sterilized in a 1% sodium hypochlorite solution (v/v) for 10 min and rinsed with sterile water three times and then soaked in sterile water at 4 °C for 24 h to adsorb water.’ Why seeds were soaked at such low temperature (4 °C) for 24h?

‘The germinated uniform seeds were transferred into plastic containers’ how old seedlings were transferred? should mention.

¼ strength of Hoagland solution was applied for how many days and when full strength was started? Should be clear

‘The dry plant tissues were mixed with HNO3 and HClO4 (v/v, 83/17) for 24 h.’ how 83/17 ratio was maintained and it’s better to simply by division.

Protein contents were measured by Bradford method but protein content data is not provided in the ms.

Please provide reference for RNA extraction and analysis methods used.

‘These data suggest that Cd had an influence on photosynthesis in the maize seedlings and had more effect on Cd-sensitive variety than Cd-tolerant variety.’ Please specify positive or negative influence?

Somewhere used ‘sugar contents’ while somewhere ‘sugar concentration’ please be specific throughout ms

‘The expression of ZmSUT1 was higher in the shoots and roots of Cd-treated FY9, and expressions of ZmSUT1 and ZmSUT4 were higher in the shoots and roots of Cd-treated SY33, than in the control (Fig 6A, B, C, D).’ But ZmSUT1 expression was higher in shoots while decreased in roots in Cd-treated FY9 as compared to control. Please confirm this result and correct.

It was Cd concentration that was determined in shoots and root tissues but in fig legends and fig 1 it is mentioned Cd content. Please correct it throughout ms.

Please explain the full names of SPS, AI, and SS in fig 4 legends

‘The total soluble sugar contents showed a clear increase when the Cd concentrations in the roots were low in the roots of the Cd-sensitive maize, SY33.’ Please revise it.

Please simplify this sentence ‘The AI and SS-H activities were higher in the shoots of SY33 than in those of FY9, and were higher in the shoots of both plants than in the control, as the soluble sugars in the shoots of both varieties increased as the Cd concentration increased.’

I would suggest authors to thoroughly go through the whole ms to correct some typographic mistakes and take help from native speaker to improve English language. For example, there is a significant need to improve main headings and the word ‘THE’ is unnecessarily used throughout ms.

Reviewer #2: (No Response)

Reviewer #3: In the manuscript „Changes in sucrose metabolism in maize varieties with different cadmium sensitivities under cadmium stress”, the authors describe the effects of increasing cadmium concentrations on the seedlings of two different maize varieties. The topic is well presented and the experimental approach is correct. The results are properly described and are clear. However, the discussion is confusing and there is not a clear thread. Each of the results are rather discussed separately and not used to build up a round story. This is perfectly reflected in the last paragraph, where instead of summarizing the conclusion and go to the point, starts to divagate once more about the results. I strongly encourage the authors to re-write the discussion and be clear, highlight the main experiment, as shown by the title (impact of cadmium on sucrose metabolism) and use the rest to support it.

7. PLOS authors have the option to publish the peer review history of their article (what does this mean?). If published, this will include your full peer review and any attached files.

Reviewer #1: **Yes: **Muhammad Imran

Reviewer #2: **Yes: **David W.M. Leung

Reviewer #3: No

---

## [Author Response · Author response to Decision Letter 1]

2 Nov 2020

Dear Editor:

We thank the editor and reviewers for comments and suggestions about our article entitled “Changes in sucrose metabolism in maize varieties with different cadmium sensitivities under cadmium stress” (ID: PONE-D-20-20550). We have revised the manuscript according to your suggestions. The revised parts are marked in red in the paper. We have compiled the main corrections and the responses to the reviewers’ comments as below.

Reviewer #1: It is appreciated that authors have tried their best to improve the manuscript but still following are some concerns that must be addressed:

1.There are no line numbers in the present form of ms and so authors should use line number option as it is difficult to highlight the mistakes at specific points

Response: We apologize for the inconvenience caused by the omission. Line number have been added to the manuscript.

2.‘The seeds were sterilized in a 1% sodium hypochlorite solution (v/v) for 10 min and rinsed with sterile water three times and then soaked in sterile water at 4 °C for 24 h to adsorb water.’ Why seeds were soaked at such low temperature (4 °C) for 24h?

Response: In this experiment we thought that seeds imbibition in low temperature could increase the consistency of seed germination. But our other experiments showed that the uniform of seed germination in low temperature was similar to that in other temperatures, such as 20 or 28°C.

3.‘The germinated uniform seeds were transferred into plastic containers’ how old seedlings were transferred? should mention.

Response: We apologize for this omission. The seedlings germinated for 3 days were transferred into plastic containers. We have added the information in marked version line 103-104.

4. ¼ strength of Hoagland solution was applied for how many days and when full strength was started? Should be clear

Response: In our previous experiment, the 1/4 Hoagland solution could provide sufficient nutrients for seedling growth at the stage of this experiment. Therefore, it was applied for the whole cultivating and stress treatment stages.

5.‘The dry plant tissues were mixed with HNO3 and HClO4 (v/v, 83/17) for 24 h.’ how 83/17 ratio was maintained and it’s better to simply by division.

Response: We improved the Cd concentrations determination method according to the method of Guo et al (2019). The ratio of HNO3 and HClO4 (v/v, 83/17) is suitable for the extraction of Cd in our experiment.

6. Protein contents were measured by Bradford method but protein content data is not provided in the ms.

Response: Thank you for your comment. The content about protein content measure was removed from ms.

7. Please provide reference for RNA extraction and analysis methods used.

Response: RNA extraction was made using RNAprep pure Plant Kit (Qiagen). The reference for analysis method has been added in marked version line 163.

8. ‘These data suggest that Cd had an influence on photosynthesis in the maize seedlings and had more effect on Cd-sensitive variety than Cd-tolerant variety.’ Please specify positive or negative influence?

Response: Thank you for your suggestion. We have added the word “negative” in marked version line 201.

9. Somewhere used ‘sugar contents’ while somewhere ‘sugar concentration’ please be specific throughout ms

Response: Thank you for your suggestion. We have replaced “sugar concentration” with “sugar contents” in marked version line 207.

10.‘The expression of ZmSUT1 was higher in the shoots and roots of Cd-treated FY9, and expressions of ZmSUT1 and ZmSUT4 were higher in the shoots and roots of Cd-treated SY33, than in the control (Fig 6A, B, C, D).’ But ZmSUT1 expression was higher in shoots while decreased in roots in Cd-treated FY9 as compared to control. Please confirm this result and correct.

Response: We apologize for the inaccurate statement. The sentence has been modified to “The expression of ZmSUT1 was higher in the shoots under Cd-treated, while it was lower than control under 20 mg L−1 Cd-treated in the roots of FY9.The expressions of ZmSUT1 and ZmSUT4 were higher in the shoots and roots of Cd-treated SY33, than in the control (Fig 6A, B, C, D).” in the line marked version 253-255.

11. It was Cd concentration that was determined in shoots and root tissues but in fig legends and fig 1 it is mentioned Cd content. Please correct it throughout ms.

Response: Thank you for your suggestion. We have replaced “Cd content” with “Cd concentration” in marked version line 26, 383, 590, and Fig. 1, respectively. 

12. Please explain the full names of SPS, AI, and SS in fig 4 legends

Response: Thank you for your suggestion. The full names of SPS, AI, and SS have been added to Fig. 4 legends in marked version line 606-607.

13.‘The total soluble sugar contents showed a clear increase when the Cd concentrations in the roots were low in the roots of the Cd-sensitive maize, SY33.’ Please revise it.

Response: Thank you for your suggestion. The sentence has been modified to “The total soluble sugar contents showed a clear increase when Cd concentrations were low in the roots of the Cd-sensitive maize, SY33.” In marked version line 321-322.

14. Please simplify this sentence ‘The AI and SS-H activities were higher in the shoots of SY33 than in those of FY9, and were higher in the shoots of both plants than in the control, as the soluble sugars in the shoots of both varieties increased as the Cd concentration increased.’

Response: Thank you for your suggestion. The sentence has been modified to “The AI and SS-H activities were higher in the shoots than in the control, and were higher in SY33 than that in FY9 as Cd concentration increased.” in marked version line 299-300.

15. I would suggest authors to thoroughly go through the whole ms to correct some typographic mistakes and take help from native speaker to improve English language. For example, there is a significant need to improve main headings and the word ‘THE’ is unnecessarily used throughout ms.

Response: Thank you for your suggestion. We have modified and improved the English language in the manuscript.

Reviewer #2: (No Response)

Reviewer #3: In the manuscript „Changes in sucrose metabolism in maize varieties with different cadmium sensitivities under cadmium stress”, the authors describe the effects of increasing cadmium concentrations on the seedlings of two different maize varieties. The topic is well presented and the experimental approach is correct. The results are properly described and are clear. However, the discussion is confusing and there is not a clear thread. Each of the results are rather discussed separately and not used to build up a round story. This is perfectly reflected in the last paragraph, where instead of summarizing the conclusion and go to the point, starts to divagate once more about the results. I strongly encourage the authors to re-write the discussion and be clear, highlight the main experiment, as shown by the title (impact of cadmium on sucrose metabolism) and use the rest to support it.

Response: Thank you for your comment and suggestion. We have modified the discussion to make it clearer. First, we investigate the phenotype of two varieties under Cd stress, and then used the changes of sucrose metabolism in different sensitive maize varieties, including the change of photosynthesis products, sucrose metabolism enzyme activities, sugar contents and the transportation and distribution of sugars, to explain the phenotype change. In the last paragraph, we illuminated the impact of cadmium on sucrose metabolism and dissected the mechanism by which sucrose metabolism response to Cd stress in two different sensitive maize varieties.

---

## [Decision Letter · Decision Letter 2]

20 Nov 2020

PONE-D-20-20550R2

Changes in sucrose metabolism in maize varieties with different cadmium sensitivities under cadmium stress

PLOS ONE

Dear Dr. Fan,

Thank you for submitting your manuscript to PLOS ONE. After careful consideration, we feel that it has merit but does not fully meet PLOS ONE’s publication criteria as it currently stands. Therefore, we invite you to submit a revised version of the manuscript that addresses the points raised during the review process.

ACADEMIC EDITOR: The revised manuscript is much improved than the previous draft, and the authors have addressed most of the comments of reviewers. However, two reviewers have raised few minor concerns, which need to be addressed/responded prior to publication.

We look forward to receiving your revised manuscript.

Kind regards,

Saddam Hussain

Academic Editor

PLOS ONE

Reviewers' comments:

Reviewer's Responses to Questions

**Comments to the Author**

1. If the authors have adequately addressed your comments raised in a previous round of review and you feel that this manuscript is now acceptable for publication, you may indicate that here to bypass the “Comments to the Author” section, enter your conflict of interest statement in the “Confidential to Editor” section, and submit your "Accept" recommendation.

Reviewer #1: All comments have been addressed

Reviewer #3: (No Response)

2. Is the manuscript technically sound, and do the data support the conclusions?

Reviewer #1: Yes

Reviewer #3: Yes

3. Has the statistical analysis been performed appropriately and rigorously? 

Reviewer #1: (No Response)

Reviewer #3: N/A

4. Have the authors made all data underlying the findings in their manuscript fully available?

Reviewer #1: Yes

Reviewer #3: Yes

5. Is the manuscript presented in an intelligible fashion and written in standard English?

Reviewer #1: (No Response)

Reviewer #3: Yes

6. Review Comments to the Author

Reviewer #1: It is appreciated that authors have incorporated suggestions/comments in the revised MS but still my concern is about statistical analysis:

The present study is comprised of two maize varieties and different Cd concentrations, so there are two independent factors but authors report at lines 167-168 that one way ANOVA was used for analyzing the collected data. So authors should have a carefully look that whether two way ANOVA was applicable? And how the interaction effects were compared and analyzed according to present statistical analysis? Although, authors have significantly improved English language but still I would suggest improving the structure/fluency of the sentences by some experts.

Reviewer #3: The authors have re-written the discussion and now it is improved. Just some minor things:

Sentence in l. 280 needs rewording, the sentence is not very clear.

l. 259, a space after stop is missing

l. 262 Brassica rapa should be written in italics

7. PLOS authors have the option to publish the peer review history of their article (what does this mean?). If published, this will include your full peer review and any attached files.

Reviewer #1: **Yes: **MUHAMMAD IMRAN

Reviewer #3: No

---

## [Author Response · Author response to Decision Letter 2]

25 Nov 2020

Dear Editor:

We thank the reviewers for comments and suggestions about our article entitled “Changes in sucrose metabolism in maize varieties with different cadmium sensitivities under cadmium stress” (ID: PONE-D-20-20550). We have revised the manuscript according to your suggestions. The revised parts are marked in red in the paper. We have compiled the main corrections and the responses to the reviewers’ comments as below.

Reviewer #1: It is appreciated that authors have incorporated suggestions/comments in the revised MS but still my concern is about statistical analysis:

The present study is comprised of two maize varieties and different Cd concentrations, so there are two independent factors but authors report at lines 167-168 that one way ANOVA was used for analyzing the collected data. So authors should have a carefully look that whether two way ANOVA was applicable? And how the interaction effects were compared and analyzed according to present statistical analysis? Although, authors have significantly improved English language but still I would suggest improving the structure/fluency of the sentences by some experts.

Response: Thank you for your suggestions. Indeed, two-way ANOVA is more applicable for this study. We have reanalyzed the data and interaction effects in two-way ANOVA. In addition, the unclear sentence has been revised. These results have been added in modified version.

Reviewer #3: The authors have re-written the discussion and now it is improved. Just some minor things:

Sentence in l. 280 needs rewording, the sentence is not very clear.

Response: I apologize for this omission. We have reworded the sentence in modified version.

l. 259, a space after stop is missing

Response: Space has been added in modified version.

l. 262 Brassica rapa should be written in italics

Response: Brassica rapa has been written in italics in modified version.

---

## [Editor Report · Decision Letter 3]

27 Nov 2020

Changes in sucrose metabolism in maize varieties with different cadmium sensitivities under cadmium stress

PONE-D-20-20550R3

Dear Dr. Fan,

We’re pleased to inform you that your manuscript has been judged scientifically suitable for publication and will be formally accepted for publication once it meets all outstanding technical requirements.

Kind regards,

Saddam Hussain

Academic Editor

PLOS ONE
---

## [Editor Report · Acceptance letter]

3 Dec 2020

PONE-D-20-20550R3 

Changes in sucrose metabolism in maize varieties with different cadmium sensitivities under cadmium stress 

Dear Dr. Fan:

I'm pleased to inform you that your manuscript has been deemed suitable for publication in PLOS ONE. Congratulations! Your manuscript is now with our production department. 

Kind regards, 

on behalf of

Dr. Saddam Hussain 

Academic Editor

PLOS ONE